# Reducing tuberculosis transmission by genotype-based contact tracing coupled with public health containment measures: a case study during the COVID-19 pandemic in Taiwan

Yuan-Shan Chien,[1,2] Chao-Chih Lai,[3,4] Chen-Yang Hsu,[4] Yu-Chu Hsieh,[2] Shin-Yi Lin,[2] Hsiao Chi Wang,[2] Hung-pin Chen,[1,5] Tony Hsiu-His Chen,[4] Dih-Ling Luh,[1,6] Yen-Po Yeh[2,4]

**ABSTRACT**   This study aimed to estimate the effectiveness of genotype-based contact tracing coupled with public health and social containment measures (PHSMs) in reducing tuberculosis (TB) transmission during the COVID-19 pandemic. Patients suspicious of recent TB infection from index cases were traced by genotyping method between 2017 and 2021. To make allowance for TB cases attributed to reactivation, TB cases identified from the genotype-based contact tracing group were compared to those from the underlying population via the notifiable nationwide system without genotyping. The relative changes (ratios) in TB cases before and during the pandemic between the two groups were leveraged to estimate the effectiveness of PHSMs following genotype-based contact tracing, taking into account demographic features and geographic variation, with a multivariable Poisson regression model. Before the pandemic, we identified 42 of 133 (31.6%) sputum culture-positive index (SCI) patients via 344 genotype-matched clustered TB cases. During the pandemic, 11 of 70 (15.7%) SCI patients were linked to 36 clustered cases. The annual average of TB-clustered patients for the genotype-based contact tracing group decreased by 84.3%, whereas the corresponding figure for the comparator decreased by 18.5%. The adjusted relative risk of 0.19 (95% CI 0.14–0.28) gave an 81% TB transmission reduction after controlling for extraneous factors. Genotype-based contact tracing coupled with PHSMs significantly reduced TB transmission. Our findings from the pandemic period demonstrate that a molecular epidemiological approach with public health containment measures will enable a moderate-burden TB country to reach the WHO End TB targets by 2035.

**IMPORTANCE**   The extent to which COVID-19 public health and social measures reduced tuberculosis transmission remains unclear. We elucidated the recent tuberculosis infection with a novel genotype-based contact tracing from 2017 to 2021. These patients were recruited as the contact tracing group in contrast to the comparison group of tuberculosis cases from the general population via the notifiable nationwide system without genotyping. The relative changes in tuberculosis cases before and during the pandemic between the contact tracing group and the comparison group were used to estimate the effectiveness of reducing tuberculosis transmission. We found a significant 81% reduction in tuberculosis transmission during the first 2 years of the pandemic. This finding demonstrates that a molecular epidemiological approach with public health containment measures will enable a moderate-burden tuberculosis country to reach the End TB targets by 2035.

**KEYWORDS**   tuberculosis, COVID-19, pandemic, public health and social measures, transmission, contact tracing, genotyping

**Peer Reviewer** Jaime Soria, University of Kentucky, Lexington, Kentucky, USA

Address correspondence to Dih-Ling Luh, luhdihling@gmail.com, or Yen-Po Yeh, yeh.leego@gmail.com.

The authors declare no conflict of interest.

Tuberculosis (TB) remains a significant global cause of death, claiming the lives of almost 1.5 million people each year. Despite significant progress achieved through international efforts in pursuit of the World Health Organization (WHO)'s goal of eliminating TB, the emergence of the COVID-19 pandemic and its disruption to healthcare systems have exerted detrimental impacts on the current state of TB case detection and treatment. According to the WHO, there was an 18% decrease in TB case notifications in 2020, accompanied by a historic reversal of the long-term downward trend in TB-related deaths worldwide (1).

Previous research has revealed two interacting dynamics that contributed to the decline in TB notifications during the pandemic. Disruptions in TB care services led to missed or delayed diagnoses of active TB diseases, whereas COVID-19 public health and social containment measures (PHSMs), such as social distancing, mask-wearing, and mobility restrictions, potentially reduced TB transmission (2–4). The majority of studies believe that the disruptions of the TB care cascade outweigh transmission reduction during the pandemic. However, the underlying mechanisms related to these two aspects remain an ongoing subject of investigation (3–7).

There is abundant evidence supporting that the strict PHSMs effectively mitigated the spread of multiple prevalent respiratory infections, such as influenza and respiratory syncytial virus (8, 9). However, research on TB transmission has been limited and inconclusive (4–6). Cross-country analysis suggested measures like stay-at-home orders and school closures were associated with decreased community spread (5). Longitudinal studies in Indonesia and South Africa support this (10, 11), while South Korea's surge in notifications among young adults indicated a higher likelihood of transmission within households (12). Of note, these investigations relied primarily on analyzing temporal changes in TB case reporting, which was confounded by the study population's mix of TB diseases from recent and remote infection (reactivation), as only the former were directly affected by the PHSMs (13).

To distinguish between these two types of TB infections, a multidisciplinary approach combining conventional epidemiological methods with mycobacterial genetics, such as genotyping, is required (14). Nonetheless, although genotyping has been widely used to detect TB cases caused by recent transmission in a population (15), no studies currently employed this methodology for disentangling the transmission dynamics of TB during the pandemic.

For the past decade, the Changhua Public Health Bureau has combined conventional contact tracing with genotyping to proactively identify and prevent TB outbreaks. Building upon this foundation, we proposed a genotype-based contact tracing approach within the context of PHSMs in Changhua, Taiwan, with the objective of assessing how TB transmission evolved during the COVID-19 pandemic. Our approach involves the utilization of nationwide genotyping data for cluster analysis, covering the period from before the pandemic (2017–2019) to during the pandemic (2020–2021). We also investigated the change in TB notifications within the general population throughout this timeframe to establish a basis for comparison.

## MATERIALS AND METHODS

### TB surveillance in Changhua, Taiwan: integrating genotyping and contact tracing

Changhua County is in central Taiwan with a population of approximately 1.26 million people and a moderate TB burden, as indicated by a TB notification rate of 45 per 100,000 in 2019 (16). The development of genotype-based contact tracing in Changhua was based on the national TB genotyping surveillance system (NTGS), established by the central government in 2012 (17). All culture-positive *Mycobacterium tuberculosis* (MTB) isolates from local laboratories nationwide were forwarded to the National Reference Laboratory of Mycobacteriology for genotyping. These specimens were subjected to the 10-locus Mycobacterial Interspersed Repetitive Unit-Variable Number Tandem Repeat

(MIRU-VNTR) analysis added on the Restriction Fragment Length Polymorphism typing of IS6110 and spacer oligonucleotide typing (spoligotyping) (18, 19). In 2015, the NTGS was incorporated into the TB Contact Tracing System as part of its regular surveillance data collection.

When a TB cluster was detected through genotyping surveillance, the health authority initiated an epidemiological investigation to trace the source of transmission and identify potential links between previously unrecognized cases. In addition to conventional active case finding, they leveraged comprehensive location-based contact tracing of individuals with the potential of having active TB disease. Furthermore, treatment efforts focused on latent TB infection in high-risk individuals or settings to prevent the spread of TB.

## Study design and subjects

To assess the effectiveness of genotyping coupled with PHSMs in reducing TB recent infection, we compared the observed change (in ratio) in TB cases of the genotype-based contact tracing group (abbreviated as the CT group) during the pandemic relative to before the pandemic (the upper panel of Fig. 1). Note that evaluating such a relative change could also reflect the preexisting declining trend of TB incidence in Taiwan during the pandemic. More importantly, as we have to allow for all reactivated TB cases that are often routinely reported through a nationwide notifiable system in the underlying general population in Taiwan, the comparison group was formed by using annual notifiable TB cases to assess the relative change as did in the contact tracing group (the lower panel of Fig. 1). The overall study design and analysis are diagramed in Fig. 1.

## Data sources

One of the extreme risks of TB in Taiwan was observed among residents of long-term care facilities (LTCFs), accounting for 10% of reported elderly TB cases and contributing significantly to TB outbreaks in congregate settings (16, 17). We therefore enrolled the TB patients in LTCFs, reported between 1 January 2017 and 31 December 2021, as the index cases for the CT group. LTCF residents also experienced serious difficulties during the COVID-19 pandemic, with an increased risk of morbidity and rapid transmission. Rigorous infection prevention and control measures were enforced in all facilities in Taiwan. Importantly, genotype-based contact tracing for TB within these facilities remained active throughout the pandemic, ensuring the timely detection and management of potential outbreaks. Our study leveraged the concomitant features of high TB risks and intensified containment efforts during the pandemic. Recall that the main reason of utilizing the comparison group is that although older people comprised 16% of Changhua's population and represented disproportionately 70% of all reported cases

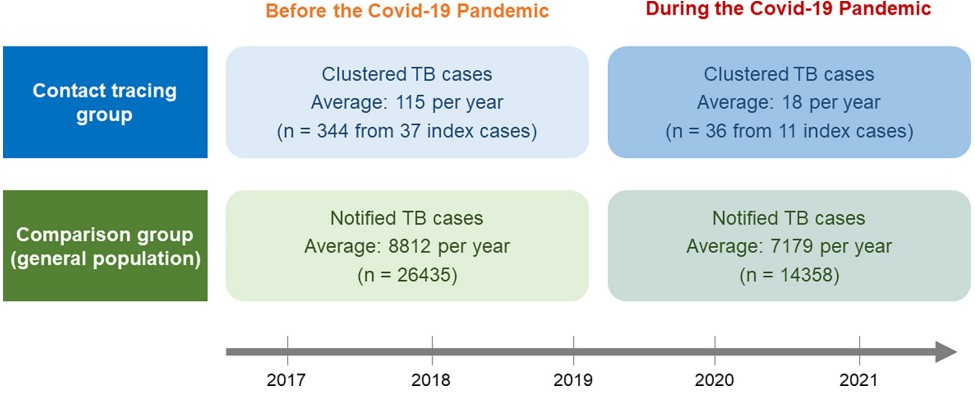

**FIG 1** The design for comparison of the changes in the number of TB cases before vs during the pandemic in the contact tracing group and the comparison group.

of TB (16), our study focused on genotype-based recent infection detection that would involve the underlying general population not limited to index cases from long-term care institutions.

The cluster was identified when two or more TB patients shared indistinguishable genotypes of MTB strains, which were reported within 2 years of the respective index case notifications. All TB cases forming clusters were included in the CT group and divided into two distinct categories based on the notification dates of the associated index cases: one before and the other during the COVID-19 pandemic (Fig. 1). Demographic characteristics and residential geographical data of each TB patient were collected for the following multivariable regression analysis.

In order to identify the comparison group (abbreviated as the CM group hereafter), we calculated the counts of reported TB patients across the nation obtained from the Taiwan National Infectious Disease Statistics System and the Taiwan TB Control Reports (16, 20). These data were stratified based on the year of notification, age, gender, and geographical areas. The categorization into periods before and during the pandemic was determined according to the date of notification.

## Statistical analyses

In order to assess whether there was a significant TB transmission reduction during the pandemic, the distribution of TB patients during the pandemic (P) relative to that before the pandemic (B) for the CT group and the CM group were compared. The ratio of P/B is a reflection of change during the pandemic compared with before the pandemic. We first calculated changes for the CT group and the CM group and then obtained the relative ratio of [(change (ratio) in the CT group) / (change (ratio) in the CM group)], denoted as RR. The magnitude of the reduction of TB transmission was represented by the value of (1-RR). Of note, RR was equivalent to the relative ratio of SMRs, calculated as $\left[ \frac{(P \text{ of the CT group}) / (P \text{ of the CM group})}{(B \text{ of the CT group}) / (B \text{ of the CM group})} \right]$. Note that P and B of the CT group can also be considered as the observed (O), and P and B of the CM group as the expected (E), which is equivalent to the relative risk of two SMR (O/E) ratios between the CT group and the CM group.

Given that the observed (O) and expected number (E) of TB cases follow a Poisson distribution, we conducted a multivariate Poisson model to evaluate the before-during associations while adjusting for gender, age, and region (21).

## Ethical aspects

During the Taipei City Hospital Institutional Review Board review process, the data collected for this study were determined to be part of program implementation and evaluation, and specific informed consent was not required by clients (TCHIRT-10611110-E). All data were de-identified to protect patient confidentiality.

## RESULTS

### The pandemic's impact on the CT group

From 2017 to 2021, a total of 261 TB patients were reported from LTCFs, with 160 instances documented before the pandemic and 101 during the pandemic, as shown in Fig. 2, demonstrating a relatively minor (5%) decline in yearly TB-notified patients. Among the 133 cases before the pandemic that tested positive for sputum culture, 42 cases (31.6%) were associated with genotype-matched clusters, resulting in the identification of 344 clustered TB cases across 30 clusters. In contrast, among the 70 patients with sputum culture positivity during the pandemic, 11 cases (15.7%) were linked to genotype-matched clusters, amounting to 36 clustered TB cases from 10 clusters overall. Overall, the proportion of index cases linked to genotype-matched clusters decreased by 50.2%, and the annual average of clustered TB patients decreased by 84.3%.

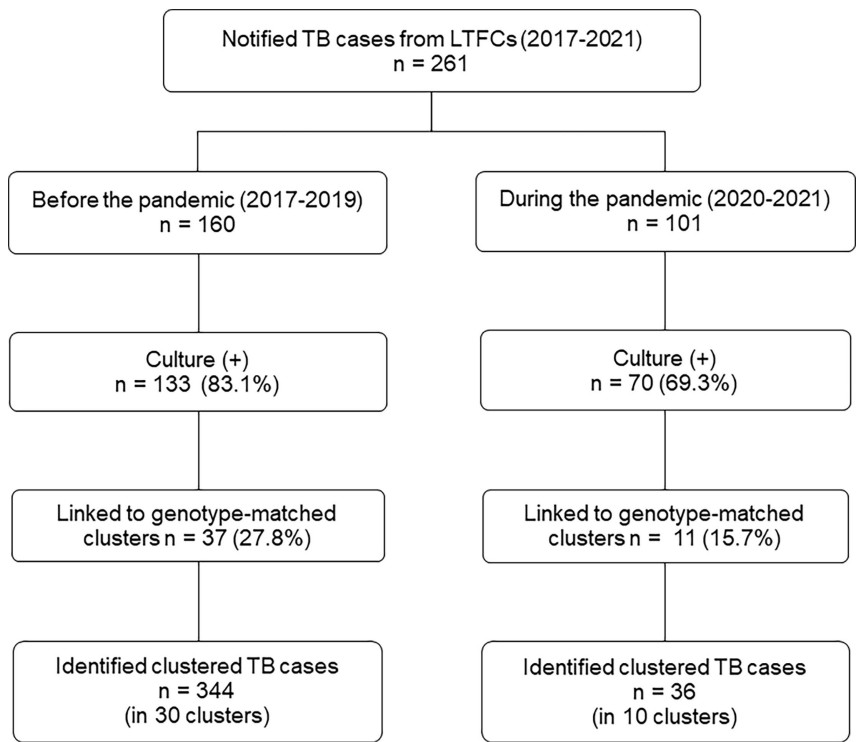

**FIG 2** Results of the genotyping-based contact tracing of the index TB cases in the long-term care facilities in Changhua, Taiwan, 2017–2021.

Figure 3 displays the geographic distribution of TB patients within clusters. Overall, noteworthy transmission links were observed across multiple locations nationwide. Before the pandemic (Fig. 3A), 308 clustered cases (89.5%) were situated outside Changhua County, encompassing nearly all other counties or cities across the country. Most of the cases occurred in metropolitan cities or urban areas. During the pandemic (Fig. 3B), the proportion of cross-jurisdictional transmission remained comparable (32 out of 36, 88.9%), yet only approximately half of the counties or cities across the country were involved, concentrated in central and northern Taiwan.

On average, during the pandemic, the occurrence of annual clustered TB cases decreased to 15.7% compared to the pre-pandemic period (Table 1). The change was more pronounced among females and younger age groups than males and older age groups. The magnitudes also varied by geographical area, fluctuating from 0% in the Kaoping region to 17.3% in the Southern region and 18.1% in the Taipei region.

## Changes in TB cases among the CM group

During the 3 years before the pandemic, there were 26,435 reported TB cases, with an annual average of 8,812 new cases. During the pandemic, however, the annual average decreased to 7,179 cases, with a total of 14,358 cases reported over a 2-year period (Table 1). The change (81.5%), which corresponded to an 18.5% decrease in TB notifications, was much less than that observed in the CT group. The level of change decreased with age, which coincided with the findings in the CT group. Furthermore, the Taipei (79.2%), Northern (78.1%), and Kaoping (81.7%) regions showed greater changes than the other regions (82.8%–90.4%). Geographically, change ratios in different regions were distributed in the opposite direction among the CM group in contrast to the CT group (Table 1; Fig. 3C and D).

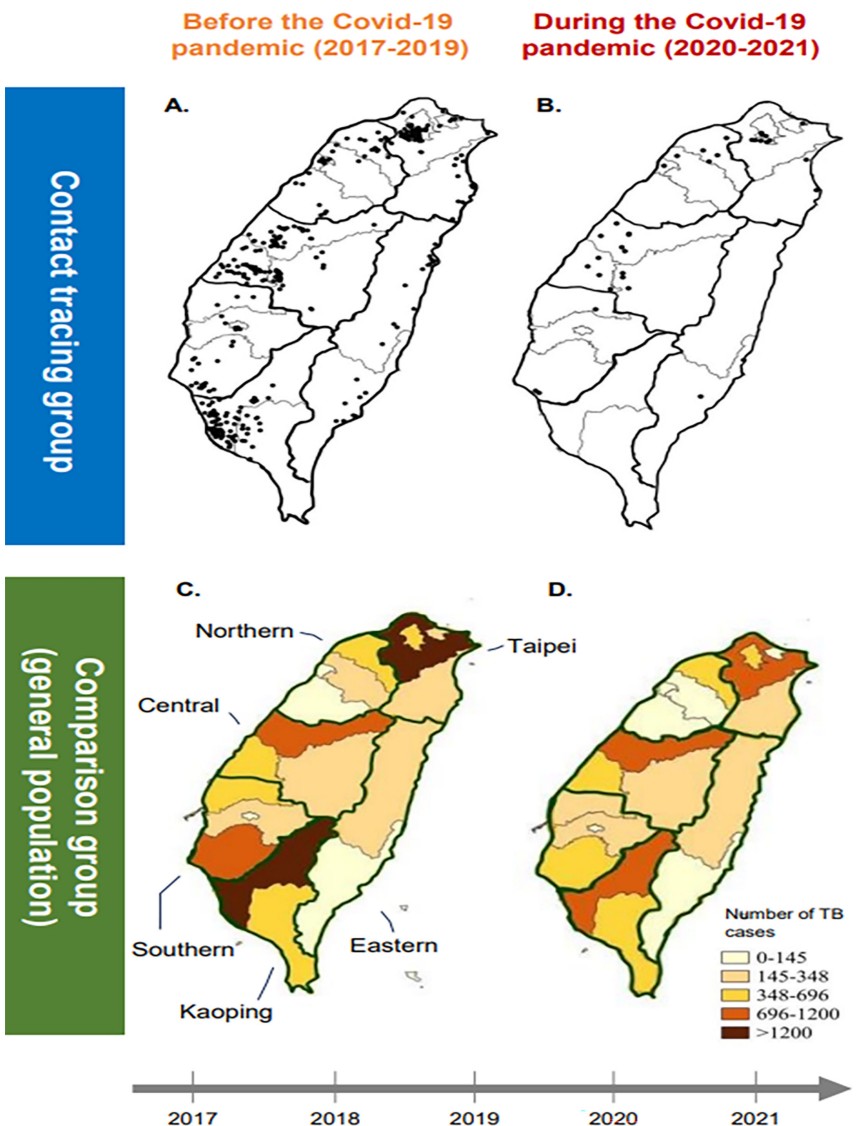

**FIG 3** Geographical distributions of genotypic clustered TB cases before (A) and during the pandemic (B) in the contact tracing group were shown, with each black dot representing one case. The distribution of the notified TB cases among the comparison group in various administrative regions was presented in different color levels. (C) Before the pandemic. (D) During the pandemic. The map was created using QGIS (https://qgis.org/) and drawn by the author.

## Relative risk of change (ratio) for the CT group vs the CM group

Overall, the RR was 0.19, giving an 80.7% reduction (Table 1). Females had lower RRs (0.15%) than males (0.21%). The younger age groups had lower RRs, with reductions ranging from 81.3% to 86.1%. Older people had the highest RR of 0.21%, corresponding to a 78.9% reduction. In comparison with other regions (0.22%–0.37%), the RRs were lower in the Southern (0.21%) and Kaoping (0%). These areas also experienced lower COVID-19 incidence and mortality (Table S1).

The multivariate Poisson model showed that the adjusted relative risk during the pandemic, compared to before, was 0.19 (95% CI, 0.14–0.28) (Table 2). This indicates a reduction of 81%, which is almost identical to the reduction derived from the relative ratio of changes (Table 1). The background transmission of TB did not differ by age group but varied in geographical regions.

**TABLE 1** Comparison of changes in the number of TB cases before the COVID-19 pandemic (2017–2019) and during the COVID-19 pandemic (2020–2021) in the contact tracing group and the comparison group, Changhua, Taiwan

| Variable | Contact tracing group[b] | | | | | Comparison group | | | | | RR (95% CI) | | Reduction (1-RR) % |
|---|---|---|---|---|---|---|---|---|---|---|---|---|---|
| | Before (B) | | During (P) | | Change[a]% | Before (B) | | During (P) | | Change[a]% | | | |
| | N | AV | N | AV | | N | AV | N | AV | | | | |
| Total | 344 | 115 | 36 | 18 | 15.7% | 26,435 | 8,812 | 14,358 | 7,179 | 81.5% | 0.19 | (0.13–0.27) | 80.7% |
| Gender | | | | | | | | | | | | | |
| Male | 255 | 85 | 29 | 15 | 17.1% | 18,379 | 6,126 | 10,051 | 5,026 | 82.0% | 0.21 | (0.14–0.31) | 79.2% |
| Female | 89 | 30 | 7 | 4 | 11.8% | 8,056 | 2,685 | 4,307 | 2,154 | 80.2% | 0.15 | (0.07–0.32) | 85.3% |
| Age (years) | | | | | | | | | | | | | |
| ≦24 | 14 | 5 | 1 | 1 | 10.7% | 992 | 331 | 379 | 190 | 57.3% | 0.19 | (0.02–1.42) | 81.3% |
| 25–44 | 28 | 9 | 2 | 1 | 10.7% | 1,750 | 583 | 897 | 449 | 76.9% | 0.14 | (0.03–0.58) | 86.1% |
| 45–64 | 101 | 34 | 9 | 5 | 13.4% | 7,768 | 2,589 | 4,064 | 2,032 | 78.5% | 0.17 | (0.09–0.34) | 83.0% |
| ≧65 | 201 | 67 | 24 | 12 | 17.9% | 15,925 | 5,308 | 9,018 | 4,509 | 84.9% | 0.21 | (0.14–0.32) | 78.9% |
| Regional | | | | | | | | | | | | | |
| Taipei | 83 | 28 | 10 | 5 | 18.1% | 7,394 | 2,465 | 3,906 | 1,953 | 79.2% | 0.23 | (0.12–0.44) | 77.2% |
| Northern | 31 | 10 | 6 | 3 | 29.0% | 3,088 | 1,029 | 1,608 | 804 | 78.1% | 0.37 | (0.16–0.89) | 62.8% |
| Central | 101 | 34 | 14 | 7 | 20.8% | 5,142 | 1,714 | 2,876 | 1,438 | 83.9% | 0.25 | (0.14–0.43) | 75.2% |
| Southern | 26 | 9 | 3 | 2 | 17.3% | 4,099 | 1,366 | 2,262 | 1,131 | 82.8% | 0.21 | (0.06–0.69) | 79.1% |
| Kaoping | 80 | 27 | 0 | 0 | 0.0% | 5,829 | 1,943 | 3,174 | 1,587 | 81.7% | 0 | | 100.0% |
| Eastern | 23 | 8 | 3 | 2 | 19.6% | 883 | 294 | 532 | 266 | 90.4% | 0.22 | (0.07–0.72) | 78.4% |

[a]The change of the number of TB cases derived from (P/B).
[b]Abbreviation: Before, before the COVID-19 pandemic; During, during the COVID-19 pandemic; TB, tuberculosis; AV, average number of TB cases per 570 year; RR, relative ratio; CI, confidence interval.

## DISCUSSION

Our investigation revealed a significant, risk-factor-adjusted reduction of 81% in TB transmissibility associated with PHSMs implemented during the initial 2 years of the pandemic, compared to a control group reflecting pre-pandemic trends and potential disruptions in TB care. While TB has a long and variable incubation period, with most infections progressing to active diseases within 2 years (13, 22), this study represents the first to directly assess recent TB infections during the pandemic. By elucidating

**TABLE 2** Results of the multivariate analysis by the Poisson model

| | Crude RR (95% CI)[a] | | Adjusted RR (95% CI)[b] | |
|---|---|---|---|---|
| COVID-19 pandemic | | | | |
| Before | 1 | | 1 | |
| During | **0.19** | **(0.13–0.27)** | **0.19** | **(0.14–0.28)** |
| Gender | | | | |
| Male | 1 | | 1 | |
| Female | **0.77** | **(0.61–0.98)** | **0.76** | **(0.63–0.96)** |
| Age (years) | | | | |
| ≦24 | 1 | | 1 | |
| 25–44 | 1.21 | (0.71–2.05) | 1.91 | (0.59–2.04) |
| 45–64 | 1.25 | (0.85–1.89) | 0.87 | (0.51–1.50) |
| ≧65 | 1.03 | (0.82–1.29) | 0.89 | (0.52–1.50) |
| Regional | | | | |
| Taipei | 1 | | 1 | |
| Northern | **0.44** | **(0.29–0.69)** | 0.94 | (0.64–1.38) |
| Central | **0.43** | **(0.25–0.71)** | **1.78** | **(1.35–2.34)** |
| Southern | 0.78 | (0.51–1.19) | **0.56** | **(0.37–0.85)** |
| Kaoping | **0.24** | **(0.15–0.42)** | 1.09 | (0.81–1.47) |
| Eastern | **0.48** | **(0.31–0.75)** | **2.27** | **(1.47–3.52)** |

[a]RR, relative risk; CI, confidence interval.
[b]Bold values denote statistical significance at the P value <0.05 level.

the underlying mechanisms of changes in TB incidence, our results bridge a crucial knowledge gap in pandemic TB epidemiology and provide robust evidence for the effectiveness of PHSMs in controlling TB.

The reduction we observed stands in stark contrast to the previously reported 10%–30% declines based on TB case notification data (1, 5). These earlier studies did not distinguish between recent infections and reactivations. The substantial discrepancy likely stems from the methodological limitations, as reactivation TB cases, unaffected by PHSMs, may have masked the actual effects of recent infections on overall trends. In the present study, the proportion of recent infection within the contact tracing group is consistent with previous research indicating that recent transmission represented 24%–32% of all TB cases in Taiwan (23, 24). Globally, studies have revealed a wide range of recent transmission proportions, varying from 7% to 72.3% across different communities (25). This heterogeneity highlights the potential variability in transmission reduction achieved by PHSMs. The absence of such context-specific insights has led several modeling studies to adopt overly conservative assumptions about transmission reduction when projecting future excess TB deaths (26). To effectively combat TB post-pandemic and inform evidence-based interventions, it is crucial to conduct in-depth studies to explore these variable effects across diverse settings and populations.

To accurately assess the pandemic's impact on TB transmission, it is essential to consider alternative explanations beyond solely attributing reduced transmission to PHSMs. A key concern is that COVID-19-related mortality among both undiagnosed and diagnosed TB patients may have reduced the number of infection sources and shortened infectious periods (4–7). This scenario is likely to occur primarily in situations with overwhelmed healthcare systems, whereas our study context differed significantly. Taiwan's comprehensive COVID-19 response strategy, implemented across the entire social environment, combined with a well-maintained healthcare delivery system, ensured minimal disruption to TB care and enabled rigorous contact tracing activities (Table S1; Fig. S1A and B) (27–30). This is evidenced by the greater reductions in TB transmission observed in regions with lower COVID-19 mortality rates, and the stable treatment outcomes for TB patients throughout the pandemic, with TB-related mortality rates declining by 11.4% (Table S2) (16). These findings strongly suggest that PHSMs played a major role in limiting TB transmission in Taiwan.

The COVID-19 pandemic presented an unprecedented natural experiment, making it possible to assess the effectiveness of PHSMs in controlling TB transmission beyond the scope of traditional healthcare-based infection prevention and control (IPC) measures. While IPC typically showed 20%–30% effectiveness (31, 32), the multifaceted PHSMs during the pandemic achieved a remarkably consistent and substantial decrease in TB transmission across diverse demographics and geographical areas. This outcome is particularly critical given the widely documented airborne transmission of TB from asymptomatic or subclinical infections in schools, congregate settings, and various community circumstances, as demonstrated by recent aerosol studies (33). In light of the resurgence of respiratory infections after PHSM relaxation (34), there is a pressing need to tailor PHSMs and scale up existing efforts to mitigate TB transmission in wider environments.

The key strength of our study is that we utilized a well-established genotype-based contact tracing approach inside the framework of PHSMs to reduce TB transmission over 2 pandemic years in a country with moderate TB prevalence. Notably, many countries with similar burdens have experienced devastating impacts from the pandemic, leading to compromised TB care and a subsequent escalation in TB transmission caused by increased undiagnosed cases (3, 5). As these countries also have a higher proportion of recent transmission TB (14, 25), they are inherently more vulnerable to PHSM effects. The extent to which the negative impacts were mitigated or exacerbated by control measures, and how these effects were modified under different social conditions, remains unclear. To explore this extremely complex scenario, the use of molecular typing technologies combined with epidemiological inquiries, as exemplified in

our work, becomes essential. However, this methodology is not yet a standard public health practice in the afflicted regions, and its adoption faces challenges due to limited resources and other pressing priorities (35). Additional support and resources are required to facilitate the deployment of this molecular epidemiological approach with interventions.

Our investigation has three limitations. First, while MIRU-VNTR typing is a well-established method for elucidating TB transmission dynamics, its utility is still constrained by the undesired specificity and positive predictive values for distinction between recent infection and reactivation (i.e., limited discriminatory power) (36, 37). This limitation is particularly relevant in endemic regions and/or areas dominated by highly conserved genotypes, such as the Beijing strains, which represented roughly half of the MTB strains in Taiwan (19, 36, 37). Additionally, the inherent potential for homoplasy, where two different strains independently evolve to have the same number of repeats at a given locus, could lead to false-positive MIRU clustering (38, 39). Nonetheless, we used the relative change (in ratio) of clustered TB cases before and during the pandemic, which may have ameliorated the influence of false-positive clustering when the comparison was made between the two periods so as not to affect our current results. Second, due to the limitations of the genotyping method's discriminatory power, the underlying pathway of transmission reduction cannot be fully explored in the current study (36, 38). Recently, high-throughput whole-genome sequencing (WGS) has emerged as a powerful tool for detecting discrete yet closely related MTB strains. Research from countries with low TB incidence has proposed that WGS can help reconstruct the putative recent transmission networks, pinpointing specific locations and individuals involved (36, 38, 40, 41). Importantly, these analyses were mostly targeted at clusters initially identified through well-established genotyping methods (40, 41). Even though there are uncertainties surrounding the effectiveness of this sequencing tool in high-incidence TB contexts and the obstacles posed by financial, technical, and infrastructural difficulties (36, 41), our findings serve to highlight the importance and potential direction of future WGS-based research endeavors. Third, the index cases in our analysis were chosen to represent Taiwan's unique epidemiological characteristics. Different index cases may have distinct contact networks, possibly resulting in diverse patterns of TB transmission across different situations. Consequently, more investigation must be done to examine these variations in variable, at-risk environments.

In conclusion, utilizing a unique genotype-based contact tracing approach in Taiwan, our study provides a comprehensive analysis of TB transmission during the initial 2 years of the pandemic. The observed significant reduction in TB transmission due to PHSMs bridges a critical gap in pandemic TB epidemiology and strongly supports their effectiveness against TB. These findings can inform the development of targeted TB control strategies post-pandemic in moderate-burden TB countries to accelerate progress toward the WHO End TB targets by 2035. Further research is essential to explore the long-term impact of PHSMs on TB transmission and optimize their implementation for global TB control.

## ACKNOWLEDGMENTS

The authors express special thanks to Professor Ruwen Jou for her invaluable assistance and insightful advice throughout this study. They are also grateful for the dedicated administrative support provided by the Reference Laboratory of Mycobacteriology, Centre for Research, Diagnostic and Vaccine Development, Center for Disease Control, Ministry of Health and Welfare of Taiwan, and the Changhua Public Health Bureau.

Y.-P.Y.: Guarantor of integrity of the entire study, study concepts, study design, definition of intellectual content, supervision or mentorship. D.-L.L.: Guarantor of integrity of the entire study, study concepts, study design, definition of intellectual content, supervision or mentorship. H.-H.C.: Guarantor of integrity of the entire study, definition of intellectual content, supervision or mentorship. Y.-S.C.: Study concepts, study design, definition of intellectual content, literature research, data acquisition, data

analysis, statistical analysis. C.-C.L.: Literature research, data analysis, statistical analysis. C.-Y.H.: Literature research, data analysis, statistical analysis. S.-Y.L.: Data acquisition, data analysis, statistical analysis. Y.-C.H.: Data acquisition. H.C.W.: Data acquisition. H.-P.C.: Data analysis, statistical analysis.

Each author contributed important intellectual content during manuscript drafting or revision and agrees to be personally accountable for the individual's own contributions and to ensure that questions pertaining to the accuracy or integrity of any portion of the work, even one in which the author was not directly involved, are appropriately investigated and resolved, including with documentation in the literature if appropriate.

## AUTHOR AFFILIATIONS

[1]Department of Public Health, Chung Shan Medical University, Taichung City, Taiwan
[2]Changhua Public Health Bureau, Changhua, Taiwan
[3]Emergency Department of Taipei City Hospital, Ren-Ai Branch, Taipei City, Taiwan
[4]Graduate Institute of Epidemiology and Preventive Medicine, College of Public Health, National Taiwan University, Taipei City, Taiwan
[5]Ershuei Township Health Center, Changhua, Taiwan
[6]Department of Family and Community Medicine, Chung Shan Medical University Hospital, Taichung City, Taiwan

## AUTHOR ORCIDs

Yuan-Shan Chien http://orcid.org/0009-0001-5228-3159
Dih-Ling Luh http://orcid.org/0000-0002-3122-060X
Yen-Po Yeh http://orcid.org/0000-0002-1091-387X

## AUTHOR CONTRIBUTIONS

Yuan-Shan Chien, Conceptualization, Data curation, Formal analysis, Methodology, Writing – original draft | Chao-Chih Lai, Data curation, Formal analysis, Methodology | Chen-Yang Hsu, Data curation, Formal analysis, Investigation, Methodology | Yu-Chu Hsieh, Data curation, Formal analysis, Investigation, Methodology | Shin-Yi Lin, Data curation, Formal analysis, Investigation, Methodology | Hsiao Chi Wang, Data curation | Hung-pin Chen, Formal analysis, Methodology | Tony Hsiu-His Chen, Supervision, Writing – review and editing | Dih-Ling Luh, Supervision, Writing – original draft, Writing – review and editing | Yen-Po Yeh, Supervision, Writing – original draft, Writing – review and editing

## ADDITIONAL FILES

The following material is available online.

### Supplemental Material

**Supplemental material (Spectrum02125-24-S0001.docx).** Tables S1 and S2; Fig. S1.

### Open Peer Review

**PEER REVIEW HISTORY (review-history.pdf).** An accounting of the reviewer comments and feedback.

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
