## [Reviewer comments · Microbiology Spectrum]

Microbiology Spectrum

Reducing Tuberculosis Transmission by Genotype-based Contact Tracing Coupled with Public Health Containment Measures: A Case Study during the COVID-19 Pandemic in Taiwan

Yuan-Shan Chien, Chao-Chih Lai, Chen-Yang Hsu, Yu-Chu Hsieh, Shin-Yi Lin, Hsiao Chi Wang, Hung-pin Chen, Hsiu-Hsi Chen, Dih-Ling Luh, and Yen-Po Yeh

Corresponding Author(s): Dih-Ling Luh, Chung Shan Medical University

Review Timeline:

Submission Date:	September 13, 2024
Editorial Decision:	October 30, 2024
Revision Received:	January 8, 2025
Editorial Decision:	February 6, 2025
Revision Received:	February 11, 2025
Accepted:	February 16, 2025

Editor: Rebecca Yee

Reviewer(s): Disclosure of reviewer identity is with reference to reviewer comments included in decision letter(s). The following individuals involved in review of your submission have agreed to reveal their identity: Jaime Soria (Reviewer #1)

Transaction Report:

DOI: <https://doi.org/10.1128/spectrum.02125-24>

Re: Spectrum02125-24 (Genotype-based Contact Tracing with Public Health Containment Measures in Reducing Tuberculosis Transmission: A Case Study during the COVID-19 Pandemic in Taiwan)

Dear Dr. Dih-Ling Luh:

Thank you for the privilege of reviewing your work. Below you will find my comments, instructions from the Spectrum editorial office, and the reviewer comments.

Please thoroughly revise and edit the manuscript for grammatical errors. While the study presented here is clear, please elaborate more on how and why the comparative groups were made. It is unclear why the comparison contact tracing group was compared to the entire population when the author states the elderly only make up 17% of the population. Further justification as well as discussion of the potential bias from this study design should be elaborated in the Discussion.

Revision Guidelines

Sincerely,
Rebecca Yee
Editor
Microbiology Spectrum

Reviewer #1 (Comments for the Author):

. The limitations of MIRU-VNTR genotyping should be discussed thoroughly, including some aspects like the fact that the MIRU-

VNTR can lead to false clustering due to homoplasy, limitations related to reproducibility between different laboratories, and potential bias due to concentrating on hypervariable loci.

- The manuscript has grammatical issues.
- In the introduction, the reference number 6 is cited after reference 10.

Reviewer #2 (Comments for the Author):

Authors of this study claim that genotype contact tracing and the use of Public Health and social containment measures reduced the risk of TB transmission during the SARS-CoV-2 pandemic. Overall, the premise of this paper demonstrates how both public health measures and TB genotyping can be utilized to contain and characterize the dynamics of TB transmission, respectively. However, major aspect of the paper needs some clarification and or further explanation. It is unclear based on the authors claim, how genotype contact tracing was utilized to reduce TB transmission. The authors describe how the genotype contact tracing was used to identify index cases and clusters, however, it is not clear what role of genotype contact tracing the authors are attributing to the decrease in TB transmission, as genotyping would only be determined on TB isolate from infected persons.

Manuscript: Spectrum02125-24

Summary of Key Findings (200-250 words)

In the manuscript, Chien et al. examine the impact of public health and social containment measures implemented during the COVID-19 pandemic on tuberculosis transmission in Taiwan. The study design uses a well-established contact tracing system integrating molecular genotyping data from long-term care facilities. The study compared the tuberculosis transmission dynamics before and during the pandemic by analyzing genotype-matched clusters and tracing new tuberculosis infections. It also distinguishes between recent transmission and reactivation. The study shows a substantial reduction in the proportion of clustered cases in the contact tracing group compared to the general population. The manuscript is well-structured and addresses an important topic in public health.

Major Concerns (at most 5-6):

The limitations of MIRU-VNTR genotyping should be discussed thoroughly, including some aspects like the fact that the MIRU-VNTR can lead to false clustering due to homoplasy, limitations related to reproducibility between different laboratories, and potential bias due to concentrating on hypervariable loci.

Minor Concerns (at most 5-6 in bullet points):

- The manuscript has grammatical issues.
- In the introduction, the reference number 6 is cited after reference 10.

Confidential Comments to the Editor:

The manuscript requires a minor revision.

Authors of this study claim that genotype contact tracing and the use of Public Health and social containment measures reduced the risk of TB transmission during the SARS-CoV-2 pandemic. Overall, the premise of this paper demonstrates how both public health measures and TB genotyping can be utilized to contain and characterize the dynamics of TB transmission, respectively. However, there were some aspects of the paper that need some clarification and/or further explanation. First, it is unclear based on the authors' claim, how genotype contact tracing was utilized to reduce TB transmission. The authors describe how the genotype contact tracing was used to identify index cases, clusters and discuss the results, however, it is not clear what role of genotype contact tracing the authors are attributing to the decrease in TB transmission, as genotyping would only be determined on TB isolate from infected persons.

Overall, the study is satisfactory, written in standard English and fairly easy to comprehend. However, there are some questions. In addition to the role of genotypic contact tracing in the authors' claim, one question on study design. The contact tracing group made up of elderly patients in long-term care facilities, which the authors describe the elderly making up 70% of all reported cases of which 10% TB cases occur in congregate settings. However, it's not clear why the comparison contact tracing group was compared to the entire population when the authors state the elderly only make up 17% of the population. Seems they should also include comparison to the total elderly population and transmission rate using standard reporting.

Reply to the comments on manuscript **Genotype-based Contact Tracing with Public Health Containment Measures in Reducing Tuberculosis Transmission: A Case Study during the COVID-19 Pandemic in Taiwan (Spectrum02125-24)** by Chien et al.

Dear editor:

Thank you for the favorable response of the Microbiology Spectrum editorial office to our manuscript cited above. We appreciate the time and effort that you and the reviewers dedicated to providing feedback on our manuscript and are grateful for the insightful comments on and valuable improvements to our paper. We are now submitting our revised manuscript for your further consideration. We have revised our manuscript along the lines suggested by the editors and reviewers. In the accompanying pages, the way we dealt with the reviewers' comments and concerns is described item by item and attached to this cover letter, and the point-by-point responses to the reviewers' comments and concerns are highlighted in blue with the changes highlighted in red in the revised text. All page numbers refer to the revised manuscript file with tracked changes. We hope very much that this revised manuscript is now acceptable for publication. Thank you for your patience in processing our manuscript for publication.

Sincere yours,

Yen-Po Yeh, M.D., PhD.

Adjunct Associate Professor, National Taiwan University, College of
Public Health, Institute of Epidemiology and Preventive Medicine
Director, Changhua County Public Health Bureau, Changhua, Taiwan

TEL: 886-4-7115141 ext 279

FAX: 886-4-7114774.

Subject: Spectrum02125-24 Decision Letter

Re: Spectrum02125-24 (Genotype-based Contact Tracing with Public Health Containment Measures in Reducing Tuberculosis Transmission: A Case Study during the COVID-19 Pandemic in Taiwan)

Dear Dr. Dih-Ling Luh:

Thank you for the privilege of reviewing your work. Below you will find my comments, instructions from the Spectrum editorial office, and the reviewer comments.

Please thoroughly revise and edit the manuscript for grammatical errors. While the study presented here is clear, please elaborate more on how and why the comparative groups were made. It is unclear why the comparison contact tracing group was compared to the entire population when the author state the elderly only make up 17% of the population. Further justification as well as discussion of the potential bias from this study design should be elaborated in the Discussion.

Revision Guidelines

To submit your modified manuscript, log into the submission site at spectrum.msubmit.net/cgi-bin/main.plex. Go to Author Tasks and click the appropriate manuscript title to begin. The information you entered when you first submitted the paper will be displayed; update this as necessary. Note the following requirements:

- Upload point-by-point responses to the issues raised by the reviewers in a file named "Response to Reviewers," NOT in your cover letter.
- Upload a compare copy of the manuscript (without figures) as a "Marked-Up Manuscript" file.
- Upload a clean .DOC/.DOCX version of the revised manuscript and remove the previous version.
- Each figure must be uploaded as a separate, editable, high-resolution file (TIFF or EPS preferred), and any multipanel figures must be assembled into one file.

- Any supplemental material intended for posting by ASM should be uploaded with their legends separate from the main manuscript. You can combine all supplemental material into one file (preferred) or split it into a maximum of 10 files with all associated legends included.

For complete guidelines on revision requirements, see our Submission and Review Process webpage.

Submission of a paper that does not conform to guidelines may delay acceptance of your manuscript.

Publication Fees: For information on publication fees and which article types are subject to charges, visit our website. If your manuscript is accepted for publication and any fees apply, you will be contacted separately about payment during the production process; please follow the instructions in that e-mail. Arrangements for payment must be made before your article is published.

ASM Membership: Corresponding authors may join or renew ASM membership to obtain discounts on publication fees. Need to upgrade your membership level? Please contact Customer Service at Service@asmusa.org.

The ASM Journals program strives for constant improvement in our submission and publication process. Please tell us how we can improve your experience by taking this quick Author Survey. Thank you for submitting your paper to Spectrum.

Sincerely,

Rebecca Yee
Editor
Microbiology Spectrum

-----1. Editor/Editorial Board Comments -----

Editor/Editorial Board Comment 1.

Please thoroughly revise and edit the manuscript for grammatical errors.

Reply 1. Thank you very much for your valuable suggestion. We have thoroughly revised the manuscript and addressed all grammatical errors. Furthermore, we have made slight adjustments to the title to better reflect the main theme of the article “**Reducing Tuberculosis Transmission by Genotype-based Contact Tracing Coupled with Public Health Containment Measures: A Case Study during the COVID-19 Pandemic in Taiwan**”.

Editor/Editorial Board Comment 2.

While the study presented here is clear, please elaborate more on how and why the comparative groups were made. It is unclear why the comparison contact tracing group was compared to the entire population when the author state the elderly only make up 17% of the population. Further justification as well as discussion of the potential bias from this study design should be elaborated in the Discussion.

Reply 2. Thank you very much for your valuable suggestion.

- (1) To assess the effectiveness of genotyping coupled with PHSMs in reducing TB recent infection, we compared the observed change (in ratio) in TB cases of the genotype-based contact tracing group during the pandemic relative to before the pandemic. The comparison is shown in the upper panel of Figure 1. Of note, such a relative change could also reflect **the pre-existing declining trend of TB incidence in Taiwan during the pandemic**. As most of the TB cases in Taiwan were reactivation TB, which were minimally affected by PHSMs. We therefore used the TB cases routinely reported through a nationwide notifiable system in the underlying general population in Taiwan as the comparison group. We assess the relative change of TB cases in the comparison group as we did in the contact tracing group (the lower panel of Figure 1). Then, the relative change of the genotype-based contact tracing group was compared to the relative change of the comparison group. The overall study design and analysis are diagrammed in Figure 1. The goal of our study was to isolate the impact of PHSMs on the transmission of recent TB infections, and the comparison group allows us to account for other factors that might influence TB case numbers.
- (2) One of the extreme risks of TB in Taiwan was observed among residents of long-term care facilities (LTCFs), accounting for 10% of reported elderly TB cases and contributing significantly to TB outbreaks in congregate settings. During the COVID-19 pandemic, rigorous infection prevention and control measures were enforced in all LTCFs in Taiwan and genotype-based contact tracing for TB within these facilities remained active. Our study leveraged the concomitant features of high TB risks and intensified containment efforts during the pandemic. Our study therefore leveraged the concomitant features of high TB risks and intensified containment efforts during the pandemic, with TB cases in LTCFs serving as the index cases for the CT group. Although older people comprised 16% of Changhua’s

population and represented disproportionately 70% of all reported cases of TB, our study focused on genotype-based recent infection detection. This focus would involve the underlying general population and not be limited to index cases from long-term care institutions.

We have modified the paragraph in the **Study design and subjects** and **The contact tracing group and the control group** sections to explain how and why the comparative groups were made, so it reads:

“Study design and subjects

To assess the effectiveness of genotyping coupled with PHSMs in reducing TB recent infection, we compared the observed change (in ratio) in TB cases of the genotype-based contact tracing group (abbreviated as the CT group) during the pandemic relative to before the pandemic (the upper panel of Figure 1). Note that evaluating such a relative change could also reflect the pre-existing declining trend of TB incidence in Taiwan during the pandemic. More importantly, as we have to allow for all reactivated TB cases that are often routinely reported through a nationwide notifiable system in the underlying general population in Taiwan, the comparison group was formed by using annual notifiable TB cases to assess the relative change as did in the contact tracing group (the lower panel of Figure 1). The overall study design and analysis are diagrammed in Figure 1.

Data sources

One of the extreme risks of TB in Taiwan was observed among residents of long-term care facilities (LTCFs), accounting for 10% of reported elderly TB cases and contributing significantly to TB outbreaks in congregate settings. We therefore enrolled the TB patients in LTCFs, reported between January 1, 2017, and December 31, 2021, as the index cases for the CT group. LTCF residents also experienced serious difficulties during the COVID-19 pandemic, with an increased risk of morbidity and rapid transmission. Rigorous infection prevention and control measures were enforced in all facilities in Taiwan. Importantly, genotype-based contact tracing for TB within these facilities remained active throughout the pandemic, ensuring the timely detection and management of potential outbreaks. Our study leveraged the concomitant features of high TB risks and intensified containment efforts during the pandemic. Recall that the main reason of utilizing the comparison group is that although older people comprised 16% of Changhua’s population and represented disproportionately 70% of all reported cases of TB, our study focused on genotype-based recent infection detection that would involve the underlying general population not limited to index cases from long-term care institutions.”

Reviewer #1 (Comments for the Author):

Reviewer #1 Comment 1.

. The limitations of MIRU-VNTR genotyping should be discussed thoroughly, including some aspects like the fact that the MIRU-VNTR can lead to false clustering due to homoplasy, limitations related to reproducibility between different laboratories, and potential bias due to concentrating on hypervariable loci.

Reply 1. Thank you very much for your valuable suggestion. We have added a paragraph in the last part of the **Discussion** section to describe the limitations of MIRU-VNTR genotyping, so it reads:

“Our investigation has three limitations. First, while MIRU-VNTR typing is a well-established method for elucidating TB transmission dynamics, its utility is still constrained by the undesired specificity and positive predictive values for distinction between recent infection and reactivation (i.e., limited discriminatory power). This limitation is particularly relevant in endemic regions and/or areas dominated by highly conserved genotypes, such as the Beijing strains, which represented roughly half of the MTB strains in Taiwan. Additionally, the inherent potential for homoplasy, where two different strains independently evolve to have the same number of repeats at a given locus, could lead to false-positive MIRU clustering. Nonetheless, we used the relative change (in ratio) of clustered TB cases before and during the pandemic, which may have ameliorated the influence of false-positive clustering when the comparison made between the two periods so as not to affect our current results.”

Reviewer #1 Comment 2.

- The manuscript has grammatical issues.

Reply 2. Thank you very much for your valuable suggestion. We have thoroughly revised the manuscript and addressed all grammatical errors.

Reviewer #1 Comment 2.

- In the introduction, the reference number 6 is cited after reference 10.

Reply 3. Thank you very much for your valuable suggestion. We have corrected the reference number.

.....

Reviewer #2 (Comments for the Author):

Reviewer #2 Comment 1.

Authors of this study claim that genotype contact tracing and the use of Public Health and social containment measures reduced the risk of TB transmission during the SARS-CoV-2 pandemic. Overall, the premise of this paper demonstrates how both public health measures and TB genotyping can be utilized to contain and characterize the dynamics of TB transmission, respectively. However, major aspect of the paper needs some clarification and or further explanation. It is unclear based on the authors claim, how genotype contact tracing was utilized to reduce TB transmission. The authors describe how the genotype contact tracing was used to identify index cases and clusters, however, it is not clear what role of genotype contact tracing the authors are attributing to the decrease in TB transmission, as genotyping would only be determined on TB isolate from infected persons.

Reply 1. Thank you very much for your valuable suggestion. We have added a paragraph in the section **TB surveillance in Changhua, Taiwan: integrating genotyping and contact tracing** to describe how the genotype contact tracing attributing to the decrease in TB transmission, so it reads:

“TB surveillance in Changhua, Taiwan: integrating genotyping and contact tracing

Changhua County is in central Taiwan with a population of approximately 1.26 million people and a moderate TB burden, as indicated by a TB notification rate of 45 per 100,000 in 2019²¹. The development of genotype-based contact tracing in Changhua was based on the national TB genotyping surveillance system (NTGS), established by the central government in 2012²². All culture-positive Mycobacterium Tuberculosis (MTB) isolates from local laboratories nationwide were forwarded to the National Reference Laboratory of Mycobacteriology for genotyping. Specimens collected before 2014 were analyzed by Restriction Fragment Length Polymorphism (RFLP) of the IS6110 and spacer oligonucleotide typing (spoligotyping), while those collected after 2014 were subjected to the 10-locus MIRU-VNTR added on the RFLP-spoligotyping. In 2015, the NTGS was incorporated into the TB Contact Tracing System (TBCTS) as part of its regular surveillance data collection.

When a TB cluster was detected through genotyping surveillance, the health authority initiated an epidemiological investigation to trace the source of transmission and identify potential links between previously unrecognized cases. In addition to conventional active case-finding, they leveraged comprehensive location-based contact tracing of individuals with the potential of having active TB disease. Furthermore, treatment efforts focused on latent TB infection (LTBI) in high-risk individuals or settings to prevent the spread of TB.”

We have also modified the **Data sources** section to emphasize that the genotype-based contact tracing for TB within LTCFs remained active throughout the pandemic, so it reads:

“Data sources

One of the extreme risks of TB in Taiwan was observed among residents of long-term care facilities (LTCFs), accounting for 10% of reported elderly TB cases and contributing significantly

to TB outbreaks in congregate settings. We therefore enrolled the TB patients in LTCFs, reported between January 1, 2017, and December 31, 2021, as the index cases for the CT group. LTCF residents also experienced serious difficulties during the COVID-19 pandemic, with an increased risk of morbidity and rapid transmission. Rigorous infection prevention and control measures were enforced in all facilities in Taiwan. **Importantly, genotype-based contact tracing for TB within these facilities remained active throughout the pandemic, ensuring the timely detection and management of potential outbreaks.** Our study leveraged the concomitant features of high TB risks and intensified containment efforts during the pandemic. Recall that the main reason of utilizing the comparison group is that although older people comprised 16% of Changhua's population and represented disproportionately 70% of all reported cases of TB, our study focused on genotype-based recent infection detection that would involve the underlying general population not limited to index cases from long-term care institutions.”

Re: Spectrum02125-24R1 (Reducing Tuberculosis Transmission by Genotype-based Contact Tracing Coupled with Public Health Containment Measures: A Case Study during the COVID-19 Pandemic in Taiwan)

Dear Prof. Dih-Ling Luh:

Thank you for the privilege of reviewing your work. Below you will find my comments, instructions from the Spectrum editorial office, and the reviewer comments.

Reviewer #3:

1. The section "Taiwan's COVID-19 success and its effectiveness on TB transmission" under Methods should not be in the Methods but rather the Discussion or Results depending on what is more appropriate. If these results are not from this manuscript, then the Discussion section would be more appropriate.
2. Lines 134-137- please evaluate if this statement is relevant to this study (e.g. isolates before 2014) and remove or edit accordingly.
3. Please once again thoroughly revise the grammar for this manuscript and ensure formatting is also correct. For example, in line 132, it states "Mycobacterium Tuberculosis". It should be *Mycobacterium tuberculosis* and in italics. Please revise manuscript thoroughly.

Revision Guidelines

Sincerely,

Rebecca Yee
Editor
Microbiology Spectrum

Reply to the comments on manuscript **Genotype-based Contact Tracing with Public Health Containment Measures in Reducing Tuberculosis Transmission: A Case Study during the COVID-19 Pandemic in Taiwan (Spectrum02125-24)** by Chien et al.

Dear editor:

Thank you for the favorable response of the Microbiology Spectrum editorial office to our manuscript cited above. We appreciate the time and effort that you and the reviewers dedicated to providing feedback on our manuscript and are grateful for the insightful comments on and valuable improvements to our paper. We are now submitting our revised manuscript for your further consideration. We have revised our manuscript along the lines suggested by the editors and reviewers. In the accompanying pages, the way we dealt with the reviewers' comments and concerns is described item by item and attached to this cover letter, and the point-by-point responses to the reviewers' comments and concerns are highlighted in blue with the changes highlighted in red in the revised text. All page numbers refer to the revised manuscript file with tracked changes. We hope very much that this revised manuscript is now acceptable for publication. Thank you for your patience in processing our manuscript for publication.

Sincere yours,

Yen-Po Yeh, M.D., PhD.

Adjunct Associate Professor, National Taiwan University, College of
Public Health, Institute of Epidemiology and Preventive Medicine
Director, Changhua County Public Health Bureau, Changhua, Taiwan

TEL: 886-4-7115141 ext 279

FAX: 886-4-7114774.

Subject: Spectrum02125-24 Decision Letter

Re: Spectrum02125-24 (Genotype-based Contact Tracing with Public Health Containment Measures in Reducing Tuberculosis Transmission: A Case Study during the COVID-19 Pandemic in Taiwan)

Dear Dr. Dih-Ling Luh:

Thank you for the privilege of reviewing your work. Below you will find my comments, instructions from the Spectrum editorial office, and the reviewer comments.

Please thoroughly revise and edit the manuscript for grammatical errors. While the study presented here is clear, please elaborate more on how and why the comparative groups were made. It is unclear why the comparison contact tracing group was compared to the entire population when the author states the elderly only make up 17% of the population. Further justification as well as discussion of the potential bias from this study design should be elaborated in the Discussion.

Revision Guidelines

To submit your modified manuscript, log into the submission site at spectrum.msubmit.net/cgi-bin/main.plex. Go to Author Tasks and click the appropriate manuscript title to begin. The information you entered when you first submitted the paper will be displayed; update this as necessary. Note the following requirements:

- Upload point-by-point responses to the issues raised by the reviewers in a file named "Response to Reviewers," NOT in your cover letter.
- Upload a compare copy of the manuscript (without figures) as a "Marked-Up Manuscript" file.
- Upload a clean .DOC/.DOCX version of the revised manuscript and remove the previous version.
- Each figure must be uploaded as a separate, editable, high-resolution file (TIFF or EPS preferred), and any multipanel figures must be assembled into one file.

- Any supplemental material intended for posting by ASM should be uploaded with their legends separate from the main manuscript. You can combine all supplemental material into one file (preferred) or split it into a maximum of 10 files with all associated legends included.

For complete guidelines on revision requirements, see our Submission and Review Process webpage.

Submission of a paper that does not conform to guidelines may delay acceptance of your manuscript.

Publication Fees: For information on publication fees and which article types are subject to charges, visit our website. If your manuscript is accepted for publication and any fees apply, you will be contacted separately about payment during the production process; please follow the instructions in that e-mail. Arrangements for payment must be made before your article is published.

ASM Membership: Corresponding authors may join or renew ASM membership to obtain discounts on publication fees. Need to upgrade your membership level? Please contact Customer Service at Service@asmusa.org.

The ASM Journals program strives for constant improvement in our submission and publication process. Please tell us how we can improve your experience by taking this quick Author Survey. Thank you for submitting your paper to Spectrum.

Sincerely,

Rebecca Yee
Editor
Microbiology Spectrum

-----1. Editor/Editorial Board Comments -----

Editor/Editorial Board Comment 1.

Please thoroughly revise and edit the manuscript for grammatical errors.

Reply 1. Thank you very much for your valuable suggestion. We have thoroughly revised the manuscript and addressed all grammatical errors. Furthermore, we have made slight adjustments to the title to better reflect the main theme of the article “Reducing Tuberculosis Transmission by Genotype-based Contact Tracing Coupled with Public Health Containment Measures: A Case Study during the COVID-19 Pandemic in Taiwan”.

Editor/Editorial Board Comment 2.

While the study presented here is clear, please elaborate more on how and why the comparative groups were made. It is unclear why the comparison contact tracing group was compared to the entire population when the author state the elderly only make up 17% of the population. Further justification as well as discussion of the potential bias from this study design should be elaborated in the Discussion.

Reply 2. Thank you very much for your valuable suggestion.

- (1) To assess the effectiveness of genotyping coupled with PHSMs in reducing TB recent infection, we compared the observed change (in ratio) in TB cases of the genotype-based contact tracing group during the pandemic relative to before the pandemic. The comparison is shown in the upper panel of Figure 1. Of note, such a relative change could also reflect **the pre-existing declining trend of TB incidence in Taiwan during the pandemic**. As most of the TB cases in Taiwan were reactivation TB, which were minimally affected by PHSMs. We therefore used the TB cases routinely reported through a nationwide notifiable system in the underlying general population in Taiwan as the comparison group. We assess the relative change of TB cases in the comparison group as we did in the contact tracing group (the lower panel of Figure 1). Then, the relative change of the genotype-based contact tracing group was compared to the relative change of the comparison group. The overall study design and analysis are diagrammed in Figure 1. The goal of our study was to isolate the impact of PHSMs on the transmission of recent TB infections, and the comparison group allows us to account for other factors that might influence TB case numbers.
- (2) One of the extreme risks of TB in Taiwan was observed among residents of long-term care facilities (LTCFs), accounting for 10% of reported elderly TB cases and contributing significantly to TB outbreaks in congregate settings. During the COVID-19 pandemic, rigorous infection prevention and control measures were enforced in all LTCFs in Taiwan and genotype-based contact tracing for TB within these facilities remained active. Our study leveraged the concomitant features of high TB risks and intensified containment efforts during the pandemic. Our study therefore leveraged the concomitant features of high TB risks and intensified containment efforts during the pandemic, with TB cases in LTCFs serving as the index cases for the CT group. Although older people comprised 16% of Changhua’s

population and represented disproportionately 70% of all reported cases of TB, our study focused on genotype-based recent infection detection. This focus would involve the underlying general population and not be limited to index cases from long-term care institutions.

We have modified the paragraph in the **Study design and subjects** and **The contact tracing group and the control group** sections to explain how and why the comparative groups were made, so it reads:

“Study design and subjects

To assess the effectiveness of genotyping coupled with PHSMs in reducing TB recent infection, we compared the observed change (in ratio) in TB cases of the genotype-based contact tracing group (abbreviated as the CT group) during the pandemic relative to before the pandemic (the upper panel of Figure 1). Note that evaluating such a relative change could also reflect the pre-existing declining trend of TB incidence in Taiwan during the pandemic. More importantly, as we have to allow for all reactivated TB cases that are often routinely reported through a nationwide notifiable system in the underlying general population in Taiwan, the comparison group was formed by using annual notifiable TB cases to assess the relative change as did in the contact tracing group (the lower panel of Figure 1). The overall study design and analysis are diagrammed in Figure 1.

Data sources

One of the extreme risks of TB in Taiwan was observed among residents of long-term care facilities (LTCFs), accounting for 10% of reported elderly TB cases and contributing significantly to TB outbreaks in congregate settings. We therefore enrolled the TB patients in LTCFs, reported between January 1, 2017, and December 31, 2021, as the index cases for the CT group. LTCF residents also experienced serious difficulties during the COVID-19 pandemic, with an increased risk of morbidity and rapid transmission. Rigorous infection prevention and control measures were enforced in all facilities in Taiwan. Importantly, genotype-based contact tracing for TB within these facilities remained active throughout the pandemic, ensuring the timely detection and management of potential outbreaks. Our study leveraged the concomitant features of high TB risks and intensified containment efforts during the pandemic. Recall that the main reason of utilizing the comparison group is that although older people comprised 16% of Changhua’s population and represented disproportionately 70% of all reported cases of TB, our study focused on genotype-based recent infection detection that would involve the underlying general population not limited to index cases from long-term care institutions.”

Reviewer #1 (Comments for the Author):

Reviewer #1 Comment 1.

. The limitations of MIRU-VNTR genotyping should be discussed thoroughly, including some aspects like the fact that the MIRU-VNTR can lead to false clustering due to homoplasy, limitations related to reproducibility between different laboratories, and potential bias due to concentrating on hypervariable loci.

Reply 1. Thank you very much for your valuable suggestion. We have added a paragraph in the last part of the **Discussion** section to describe the limitations of MIRU-VNTR genotyping, so it reads:

“Our investigation has three limitations. First, while MIRU-VNTR typing is a well-established method for elucidating TB transmission dynamics, its utility is still constrained by the undesired specificity and positive predictive values for distinction between recent infection and reactivation (i.e., limited discriminatory power). This limitation is particularly relevant in endemic regions and/or areas dominated by highly conserved genotypes, such as the Beijing strains, which represented roughly half of the MTB strains in Taiwan. Additionally, the inherent potential for homoplasy, where two different strains independently evolve to have the same number of repeats at a given locus, could lead to false-positive MIRU clustering. Nonetheless, we used the relative change (in ratio) of clustered TB cases before and during the pandemic, which may have ameliorated the influence of false-positive clustering when the comparison made between the two periods so as not to affect our current results.”

Reviewer #1 Comment 2.

- The manuscript has grammatical issues.

Reply 2. Thank you very much for your valuable suggestion. We have thoroughly revised the manuscript and addressed all grammatical errors.

Reviewer #1 Comment 2.

- In the introduction, the reference number 6 is cited after reference 10.

Reply 3. Thank you very much for your valuable suggestion. We have corrected the reference number.

.....

Reviewer #2 (Comments for the Author):

Reviewer #2 Comment 1.

Authors of this study claim that genotype contact tracing and the use of Public Health and social containment measures reduced the risk of TB transmission during the SARS-CoV-2 pandemic. Overall, the premise of this paper demonstrates how both public health measures and TB genotyping can be utilized to contain and characterize the dynamics of TB transmission, respectively. However, major aspect of the paper needs some clarification and or further explanation. It is unclear based on the authors claim, how genotype contact tracing was utilized to reduce TB transmission. The authors describe how the genotype contact tracing was used to identify index cases and clusters, however, it is not clear what role of genotype contact tracing the authors are attributing to the decrease in TB transmission, as genotyping would only be determined on TB isolate from infected persons.

Reply 1. Thank you very much for your valuable suggestion. We have added a paragraph in the section **TB surveillance in Changhua, Taiwan: integrating genotyping and contact tracing** to describe how the genotype contact tracing attributing to the decrease in TB transmission, so it reads:

“TB surveillance in Changhua, Taiwan: integrating genotyping and contact tracing

Changhua County is in central Taiwan with a population of approximately 1.26 million people and a moderate TB burden, as indicated by a TB notification rate of 45 per 100,000 in 2019²¹. The development of genotype-based contact tracing in Changhua was based on the national TB genotyping surveillance system (NTGS), established by the central government in 2012²². All culture-positive Mycobacterium Tuberculosis (MTB) isolates from local laboratories nationwide were forwarded to the National Reference Laboratory of Mycobacteriology for genotyping. Specimens collected before 2014 were analyzed by Restriction Fragment Length Polymorphism (RFLP) of the IS6110 and spacer oligonucleotide typing (spoligotyping), while those collected after 2014 were subjected to the 10-locus MIRU-VNTR added on the RFLP-spoligotyping. In 2015, the NTGS was incorporated into the TB Contact Tracing System (TBCTS) as part of its regular surveillance data collection.

When a TB cluster was detected through genotyping surveillance, the health authority initiated an epidemiological investigation to trace the source of transmission and identify potential links between previously unrecognized cases. In addition to conventional active case-finding, they leveraged comprehensive location-based contact tracing of individuals with the potential of having active TB disease. Furthermore, treatment efforts focused on latent TB infection (LTBI) in high-risk individuals or settings to prevent the spread of TB.”

We have also modified the **Data sources** section to emphasize that the genotype-based contact tracing for TB within LTCFs remained active throughout the pandemic, so it reads:

“Data sources

One of the extreme risks of TB in Taiwan was observed among residents of long-term care facilities (LTCFs), accounting for 10% of reported elderly TB cases and contributing significantly

to TB outbreaks in congregate settings. We therefore enrolled the TB patients in LTCFs, reported between January 1, 2017, and December 31, 2021, as the index cases for the CT group. LTCF residents also experienced serious difficulties during the COVID-19 pandemic, with an increased risk of morbidity and rapid transmission. Rigorous infection prevention and control measures were enforced in all facilities in Taiwan. **Importantly, genotype-based contact tracing for TB within these facilities remained active throughout the pandemic, ensuring the timely detection and management of potential outbreaks.** Our study leveraged the concomitant features of high TB risks and intensified containment efforts during the pandemic. Recall that the main reason of utilizing the comparison group is that although older people comprised 16% of Changhua's population and represented disproportionately 70% of all reported cases of TB, our study focused on genotype-based recent infection detection that would involve the underlying general population not limited to index cases from long-term care institutions.”

Reply to the comments on revised manuscript **Reducing Tuberculosis Transmission by Genotype-based Contact Tracing Coupled with Public Health Containment Measures: A Case Study during the COVID-19 Pandemic in Taiwan (Spectrum02125-24R1) by Chien et al.**

Dear editor:

We sincerely appreciate the positive response from the Microbiology Spectrum editorial office regarding our revised manuscript. We are grateful for the time and effort that you and the reviewers have dedicated to providing thoughtful feedback, as well as for the insightful comments and suggestions that have helped improve our work. We are pleased to submit our revised manuscript for your further consideration. In this revision, we have carefully addressed the concerns raised by the editors and reviewers. A detailed point-by-point response to the reviewers' comments is provided in the accompanying document, where **our responses are highlighted in blue**, and **the corresponding revisions in the manuscript are marked in red**. All page numbers referenced in our response correspond to the revised manuscript file with tracked changes. We sincerely hope that this revised version meets the journal's expectations and is now suitable for publication. Thank you for your time and patience in handling our manuscript. We greatly appreciate your consideration.

Sincere yours,

Yen-Po Yeh, M.D., PhD.

Adjunct Associate Professor, National Taiwan University, College of Public Health, Institute of Epidemiology and Preventive Medicine
Director, Changhua County Public Health Bureau, Changhua, Taiwan

TEL: 886-4-7115141 ext 279

FAX: 886-4-7114774.

Re: Spectrum02125-24R1 (Reducing Tuberculosis Transmission by Genotype-based Contact Tracing Coupled with Public Health Containment Measures: A Case Study during the COVID-19 Pandemic in Taiwan)

Dear Prof. Dih-Ling Luh:

Thank you for the privilege of reviewing your work. Below you will find my comments, instructions from the Spectrum editorial office, and the reviewer comments.

Reviewer #3:

1. The section "Taiwan's COVID-19 success and its effectiveness on TB transmission" under Methods should not be in the Methods but rather the Discussion or Results depending on what is more appropriate. If these results are not from this manuscript, then the Discussion section would be more appropriate.
2. Lines 134-137- please evaluate if this statement is relevant to this study (e.g. isolates before 2014) and remove or edit accordingly.
3. Please once again thoroughly revise the grammar for this manuscript and ensure formatting is also correct. For example, in line 132, it states "Mycobacterium Tuberculosis". It should be *Mycobacterium tuberculosis* and in italics. Please revise manuscript thoroughly.

Revision Guidelines

- Upload point-by-point responses to the issues raised by the reviewers in a file named "Response to Reviewers," NOT in your cover letter.
- Upload a compare copy of the manuscript (without figures) as a "Marked-Up Manuscript" file.
- Upload a clean .DOC/.DOCX version of the revised manuscript and remove the previous version.
- Each figure must be uploaded as a separate, editable, high-resolution file (TIFF or EPS preferred), and any multipanel figures must be assembled into one file.

- Any supplemental material intended for posting by ASM should be uploaded with their legends separate from the main manuscript. You can combine all supplemental material into one file (preferred) or split it into a maximum of 10 files with all associated legends included.

Publication Fees: For information on publication fees and which article types are subject to charges, visit our website. If your manuscript is accepted for publication and any fees apply, you will be contacted separately about payment during the production process; please follow the instructions in that e-mail. Arrangements for payment must be made before your article is published.

ASM Membership: Corresponding authors may join or renew ASM membership to obtain discounts on publication fees. Need to upgrade your membership level? Please contact Customer Service at Service@asmusa.org.

Sincerely,
Rebecca Yee
Editor
Microbiology Spectrum

-----REVIEWER COMMENTS-----

Reviewer #3 (Comments for the Author):

Reviewer #3: Comment 1.

1. The section "Taiwan's COVID-19 success and its effectiveness on TB transmission" under Methods should not be in the Methods but rather the Discussion or Results depending on what is more appropriate. If these results are not from this manuscript, then the Discussion section would be more appropriate.

Reply 1. Thank you very much for your valuable suggestion. We have removed this section. Taiwan's COVID-19 success and its effectiveness on TB transmission have been described in the Discussion section.

Reviewer #3: Comment 2.

2. Lines 134-137- please evaluate if this statement is relevant to this study (e.g. isolates before 2014) and remove or edit accordingly.

Reply 1. Thank you very much for your valuable suggestion. We have removed the statement concerning the isolates before 2014 and modified this paragraph, so it reads:

“All culture-positive *Mycobacterium tuberculosis* (MTB) isolates from local laboratories nationwide were forwarded to the National Reference Laboratory of Mycobacteriology for genotyping. These specimens were subjected to the 10-locus Mycobacterial Interspersed Repetitive Unit–Variable Number Tandem Repeat (MIRU-VNTR) analysis added on the Restriction Fragment Length Polymorphism (RFLP) typing of IS6110 and spacer oligonucleotide typing (spoligotyping).”

Reviewer #3: Comment 3.

3. Please once again thoroughly revise the grammar for this manuscript and ensure formatting is also correct. For example, in line 132, it states "Mycobacterium Tuberculosis". It should be *Mycobacterium tuberculosis* and in italics. Please revise manuscript thoroughly.

Reply 1. Thank you very much for your valuable suggestion. We have meticulously revised the grammar of this manuscript and verified the formatting is accurate. The sentence now reads as follows:

“All culture-positive *Mycobacterium tuberculosis* (MTB) isolates from local laboratories nationwide were forwarded to the National Reference Laboratory of Mycobacteriology for genotyping.”

Re: Spectrum02125-24R2 (Reducing Tuberculosis Transmission by Genotype-based Contact Tracing Coupled with Public Health Containment Measures: A Case Study during the COVID-19 Pandemic in Taiwan)

Dear Prof. Dih-Ling Luh:

Your manuscript has been accepted, and I am forwarding it to the ASM production staff for publication. Your paper will first be checked to make sure all elements meet the technical requirements. ASM staff will contact you if anything needs to be revised before copyediting and production can begin. Otherwise, you will be notified when your proofs are ready to be viewed.

Sincerely,
Rebecca Yee
Editor
Microbiology Spectrum